

# Seasonally frugivorous forest birds and window collision fatalities: novel integration of bird counts in fall improves assessment of species vulnerability to collisions

Viviane Zulian[1,*], Louise K. Blight[2,3], Jon Osborne[4], Adam C. Smith[5], Andrea R. Norris[6], Rebecca Golat[7] and Krista L. De Groot[8,*]

[1] Department of Biological Sciences, Clemson University, Clemson, SC, United States of America
[2] School of Environmental Studies, University of Victoria, Victoria, British Columbia, Canada
[3] Procellaria Research & Consulting, Salt Spring Island, British Columbia, Canada
[4] Sanicle Environmental Consulting, Victoria, British Columbia, Canada
[5] Canadian Wildlife Service, Environment and Climate Change Canada, Ottawa, Ontario, Canada
[6] Science and Technology Branch, Environment and Climate Change Canada, Nanaimo, British Columbia, Canada
[7] University of Victoria, Victoria, British Columbia, Canada
[8] Science and Technology Branch, Environment and Climate Change Canada, Delta, British Columbia, Canada
[*] These authors contributed equally to this work.

Corresponding author
Krista L. De Groot,
krista.degroot@ec.gc.ca

## ABSTRACT

Bird-window collisions are a significant and growing threat to birds, but the issue is still understudied in many geographical areas and stages of the avian annual life cycle. The mountainous topography and numerous distinct biogeoclimatic zones along the Pacific coast of Canada and the United States may result in regional and seasonal differences in collision mortality and species vulnerability to collisions. We surveyed daily for evidence of bird-window collisions over six 21-day periods in fall, early winter, and late winter between 2019 and 2022 at a university campus in southwestern British Columbia, Canada, and assessed individual species' vulnerability to collisions by examining whether species-specific collision rates were disproportionate to their local abundance. We accounted for poor detectability of some species in fall, by integrating point count data from our study site with mist net capture data from a nearby banding station to improve abundance estimates. Collision mortality peaked in fall, but early winter collision mortality was significantly higher than in the later winter months, potentially due to movements of altitudinal migrants into our low-elevation study area in early winter. We estimated that an average of 885–1,342 (median = 1,095) birds are killed at 51 buildings campus-wide each year between September 15 and February 10, the peak fall migration wintering period. Forest birds, particularly species that switch to highly frugivorous diets in fall and winter, were most vulnerable to collisions across the seasons studied. Non-breeding season mortality due to collisions may be substantial for these species, particularly when considering cumulative mortality across the entire non-breeding period. The potential role of collision mortality in species declines should be further explored by assessing collision frequency and species vulnerability across life cycle stages in other geographical locations, and through improved data on migratory

connectivity and linkages between declining breeding populations and non-breeding season mortality.

## INTRODUCTION

Urban environments provide habitat for a diverse array of species, including more than 20% of the world's birds (*Aronson et al., 2014*; *Lepczyk, Aronson & La Sorte, 2023*; *Stanford et al., 2025*). While a number of bird species have adapted to breeding in remnant forest and other vegetated sites in urban and peri-urban areas, many additional species use these habitats during the non-breeding period, as migratory stop-over sites (*Seewagen et al., 2011*; *Buler & Dawson, 2014*), for post-breeding molt (*Morales et al., 2022*; *Poirier et al., 2024*; *Poirier, Elliott & Frei, 2024*) and during the wintering period of temperate-zone breeding birds (*Campbell et al., 1997*; *Campbell et al., 2001*; *Muñoz Pacheco & Villaseñor, 2022*; *Pacheco-Muñoz, Aguilar-Gómez & Schondube, 2022*). However, the built environment poses significant threats to birds globally due to the risk of collisions with glass (*Klem, 1979*; *Klem, 1990*; *Klem, 2006*; *Aymí et al., 2017*; *Hager et al., 2017*; *Basilio, Moreno & Piratelli, 2020*; *Shi et al., 2022*; *Ogłęcki & Zabicka, 2023*). Birds may be killed by flying into buildings while trying to access vegetation or sky that is visible in reflections, or on the other side of transparent glass such as glass railings and walkways (*Ross, 1945*; *Klem, 1989*). Building mortality risk varies according to a complex combination of factors, including proportion of glass area (*Klem et al., 2009*), building size (*Hager et al., 2017*), local and landscape-level vegetation surrounding the building (*Hager et al., 2017*), and number of stories reflecting vegetation: an interaction between glass height and vegetation height (*Zulian et al., 2023*). At night, migrating birds can become disoriented by artificial light and collide with tall towers or other structures in the airspace (*Adams et al., 2021*), or they can collide after daybreak as they move through the built environment (*Klem, 2006*). Building glass has been estimated to kill 16–42 million birds in Canada and 365–988 million in the United States of America (hereafter U.S.) (*Machtans, Wedeles & Bayne, 2013*; *Loss et al., 2014*); however, these figures are based on the number of buildings that existed over a decade ago. In addition, mortality and sub-lethal effects may be higher than previously estimated, as many collision events leave no measurable evidence at the site (*Samuels et al., 2022*; *Klem, Saenger & Brogle, 2024*) and recent studies reported that approximately 60% of birds that survived a collision and received rehabilitation care subsequently died (*Kornreich et al., 2024*). There is increasing urgency to address this growing conservation threat as new construction continues to accelerate globally, with building floor area projected to double by 2060 (*World Economic Forum, 2024*).

## Seasonal differences in bird collision rates

Bird-window collision studies often report peaks in mortality during the migratory periods (*Basilio, Moreno & Piratelli, 2020*). However, high mortality can also occur during the non-breeding annual life cycle stage between latitudinal fall and spring migrations, or the wintering period for temperate-zone breeding birds (hereafter referred to as "winter") (*Seavy et al., 2012*; *Horton et al., 2020*), at homes with feeders (*Dunn, 1993*; but see *O'Connell, 2001*), and in regions with high densities of resident individuals and latitudinal or altitudinal migrants (*Campbell et al., 1997*; *Campbell et al., 2001*; *Muñoz Pacheco & Villaseñor, 2022*; *Pacheco-Muñoz, Aguilar-Gómez & Schondube, 2022*). At a university campus on the Pacific coast in Canada, *De Groot et al. (2021)* found that the collision mortality rate in winter was equivalent to the rate of mortality during spring migration. Therefore, cumulative collision mortality across the five-month wintering period could surpass the total collision mortality during the shorter spring migration period in that region (*Zulian et al., 2023*). A study in coastal California, U.S., found that high numbers of collisions occurred during late summer and fall migration, continuing into November (*Kahle, Flannery & Dumbacher, 2016*), and collision mortality rate was consistent across seasons in Xalapa, Mexico (*Gómez-Martínez et al., 2019*). Timing of mortality is an important consideration, because changes in population size during one season can influence population dynamics in subsequent seasons, and density-dependent increases in productivity may not occur uniformly across all species to compensate for this mortality (*Norris & Marra, 2007*; *Calvert, Walde & Taylor, 2009*; *Longcore & Smith, 2013*; *Klaassen et al., 2014*; *Hoover et al., 2020*; *Socolar et al., 2024*).

## Species-specific vulnerability to collisions

Species' life histories may influence differential vulnerability to collision mortality and the potential for collision mortality to exert population-level effects (*Elmore et al., 2021*). Differences in migratory distance, migratory behavior, migratory altitude, migratory status, familiarity with the local environment, habitat, foraging guild, foraging strategy, flight velocity, and functional morphology, such as wing type, are all hypothesized to influence species vulnerability to collisions (*Klem, 1989*; *Arnold & Zink, 2011*; *Kahle, Flannery & Dumbacher, 2016*; *Soares Santos, de Abreu & de Vasconcelos, 2017*; *Wittig et al., 2017*; *Nichols et al., 2018*; *De Groot et al., 2021*; *Colling et al., 2022*; *Berigan et al., 2025*). In North America, New World sparrows (Passerellidae), New World warblers (Parulidae), hummingbirds (Trochillidae), and thrushes (Turdidae) appear to be more vulnerable to collisions than other family groups (*Machtans, Wedeles & Bayne, 2013*; *Loss et al., 2014*; *Kahle, Flannery & Dumbacher, 2016*; *Rebolo-Ifrán, Di Virgilio & Lambertucci, 2019*; *Elmore et al., 2021*). Similar findings are reported for thrushes and hummingbirds in some tropical regions (*Menacho-Odio, 2015*; *Soares Santos, de Abreu & de Vasconcelos, 2017*; *Gómez-Martínez et al., 2019*; *Fornazari et al., 2021*). However, considerable variation exists at the species- and genus-level within these higher order taxonomic groups (*Colling et al., 2022*). Moreover, seasonal or regional differences in diet and habitat use, building features, and landscape- and local-scale vegetation can also create geographical and temporal variation

in species' vulnerability to collisions (*Cusa, Jackson & Mesure, 2015*; *Gómez-Martínez et al., 2019*; *Riding, O'Connell & Loss, 2020*; *De Groot et al., 2021*; *Fornazari et al., 2021*).

Assessment of species vulnerability requires both collision counts and information on the relative abundance of species at a local level, thereby highlighting species that collide more frequently or less frequently than expected (*Arnold & Zink, 2011*). Estimating abundance during migration is particularly challenging, given the high motility of individuals and large differences in detectability among species (*Bennett et al., 2024*). Protocols and modeling approaches to overcome biases associated with point counts for breeding season surveys (*Nichols et al., 2000*; *Farnsworth et al., 2005*; *Sauer & Link, 2011*) are not designed to deal with the unique detectability challenges, *e.g.*, of molt migrants and other inconspicuous species, during the migratory period (*Tonra & Reudink, 2018*; *Brunner et al., 2022*). *Colling et al. (2022)* used catch data from both a nearby and a regional banding station to estimate local abundance; however, they derived independent indices of vulnerability to collisions, or 'catch ratios', *versus* integrating site-specific count data with mist net capture data. To address the issue of species detectability biasing estimates of species' vulnerability to collisions, we devised a novel analytical method to integrate weekly point count data collected at our study buildings with mist net capture data from a local banding station to improve species' abundance estimates.

Studies from the Pacific coastal regions of Canada and the U.S., which support high densities of passerines throughout the non-breeding period, are still underrepresented in the collision literature compared to studies in these countries' eastern temperate regions (*Loss et al., 2014*; but see *Kahle, Flannery & Dumbacher, 2016*; *De Groot et al., 2021*; *Zulian et al., 2023*). Data on seasonal differences in mortality and species vulnerability to collisions across a broader range of geographical locations will improve demographic models, and allow governments to adapt policies, migratory bird management, regulatory frameworks, environmental assessments, and spatial prioritization of conservation and collision mitigation efforts (*Loss, Will & Marra, 2012*).

We monitored bird-window collisions, performed trials of carcass persistence and searcher efficiency, and conducted point counts during fall migration, the early wintering stage (hereafter early winter), and the late wintering stage (hereafter late winter) in our region, on a university campus situated in a suburban area along the Pacific coast of North America; this region supports high bird densities throughout the non-breeding season. We collected these data over four years and (a) estimated season-specific collision mortality, while correcting for biases that affect the detection of carcasses; (b) estimated campus-wide mortality across these periods to better quantify the potential effects of mortality that occur at the campus during the fall and winter periods; and (c) assessed species-specific vulnerability to bird-window collisions using abundance estimates derived through integration of point counts and mist net capture data.

## MATERIALS AND METHODS
### Study site and building selection
We conducted our bird collision research at the 160 ha University of Victoria campus (48°27′47.2800N, 123°18′44.8300W, elevation 60 m) situated within a suburb of Victoria,

British Columbia (hereafter, B.C.), Canada (Fig. 1) on the unceded traditional territory of the ləkʷəŋən (Songhees and Xʷsepsəm Nations) and the W̱SÁNEĆ Peoples (*NativeLand Digital, 2025*). The campus consists of low- and mid-rise institutional buildings, surrounded by extensive native and non-native ornamental plantings, and considerable remnant second-growth forest (*i.e.,* native forest regrown following historical logging). Since building size can influence collision mortality rates on a per building basis (*Hager et al., 2017*; *Machtans, Wedeles & Bayne, 2013*; *Loss et al., 2014*), we stratified our random design by building height to select six study buildings for the surveys (Figs. 2 and 3). This ensured that study buildings did not disproportionately include the largest buildings on campus, and in aligning study design with that of another study in the region (*De Groot et al., 2021*), we were better positioned to compare our per building collision rates to a nearby campus in coastal B.C. Following *De Groot et al. (2021)*, we sampled from all buildings > 1 story, excluding student residences and medical clinics due to privacy concerns. We also excluded windowless utility sheds and storage buildings, and houses that had been converted to administrative space because they did not resemble most institutional buildings on campus. After examining the height distribution of all remaining candidate buildings on campus, we divided them into two categories, *i.e.,* "low" (2–3 stories) and "high" (4–7 stories), and buildings were randomly chosen from a list generated for each height category. In contrast to the published study mentioned above, we did not further stratify our building sample by percent vegetation cover surrounding building perimeters. This was due to a high percentage of vegetated cover, and low variation for this variable surrounding campus buildings at the University of Victoria. We used individual façades as our survey units, and our six study buildings had a combined total of 50 façades. Eighteen of the 50 study façades (36%) had overhangs, balconies or other structures that prevented carcasses of birds that hit upper-story windows from falling to the ground where they could be observed. Therefore, we could only monitor the first story of these 18 façades.

## Data collection
### Bird-window collision surveys
We surveyed study building perimeters daily for collision evidence over two fall, one early winter, and three late winter 21-day periods over four calendar years (2019–2022), following the protocols described by *Hager & Cosentino (2014)* and *De Groot et al. (2021)*. Fall survey dates were from September 16 to October 6, 2019, and September 19 to October 10, 2020, within the peak of latitudinal songbird migratory activity for the area, based on local banding station data (*Leckie, 2008*). To allow for regional comparisons, winter survey dates were chosen to coincide with those selected by *De Groot et al. (2021)* at the University of British Columbia campus located in Vancouver, B.C., on the traditional, unceded territory of the xʷməθkʷəy̓əm (Musqueam) People (*NativeLand Digital, 2025*), 90 km to the northeast of our study site, across the Strait of Georgia. Late winter surveys were therefore conducted from January 21 to February 10, 2020, January 19 to February 8, 2021, and January 17 to February 6, 2022. To explore potential within-winter differences in collision mortality, we added an early winter survey period between November 22 to December 12, 2021.

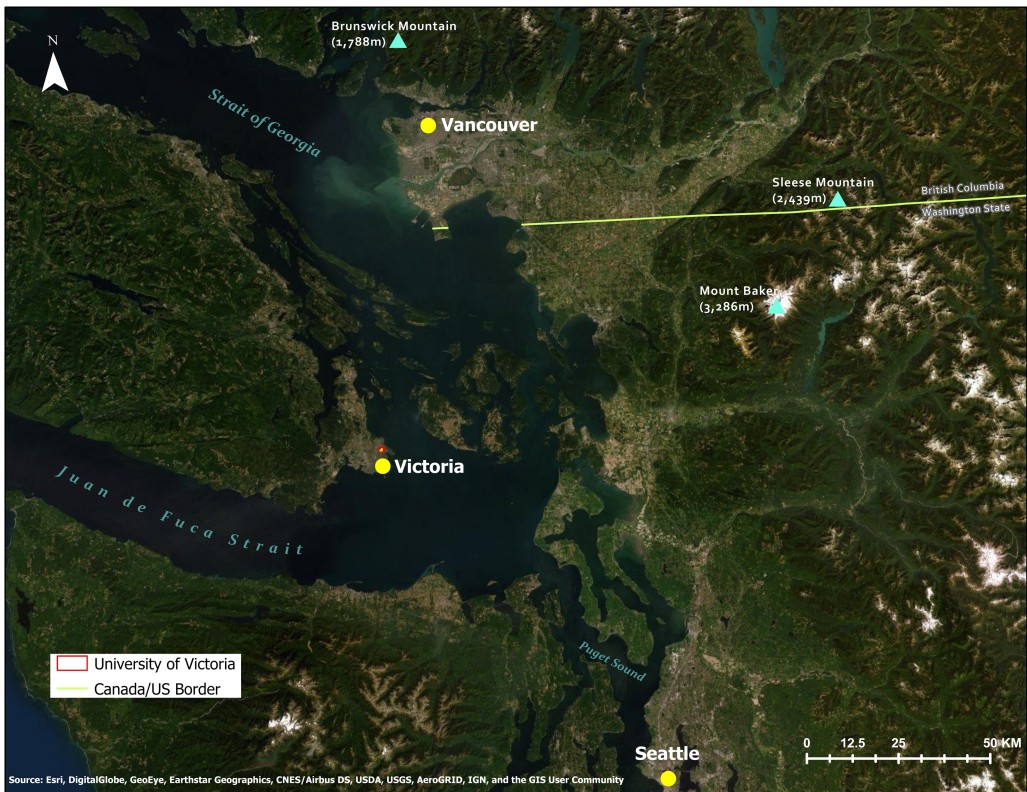

**Figure 1** **Topographical map of study area and location of study site within southern Vancouver Island, along the Pacific coast of British Columbia, Canada.** Source: Esri, Digital Globe, GeoEye, Earthstar Geographics.

On the day prior to each of these six study periods, we performed a complete clean-up of all carcasses, feather piles, and feather smears at each study building, to ensure that evidence accumulated prior to the study period was not included in the collision tallies. For each daily survey, two observers simultaneously walked the perimeter of all study buildings in opposite directions in early to mid-afternoon, searching within two meters of the façade for evidence of bird-window collisions. Individuals varied direction of their surveys and did not share information about carcasses or other evidence found until the survey was completed. Collision monitoring across the six sampling periods was carried out by a total of three observers, with one observer performing all collision surveys across all study periods, one observer participating in all surveys in the first study period, and one observer participating in all surveys in all remaining five study periods. Collision mortality evidence included all bird carcasses, as defined by *Hager & Cosentino (2014)*: intact carcasses, partially scavenged carcasses, and feather piles. We defined a feather pile as the remains of an entirely scavenged carcass with 10 or more feathers within a one meters radius, to avoid counting feathers shed naturally. We assumed that intact carcasses and remains of carcasses found within two meters of buildings were the result of birds killed outright by windows, or by predators that captured and consumed birds that were stunned by hitting windows. Additional

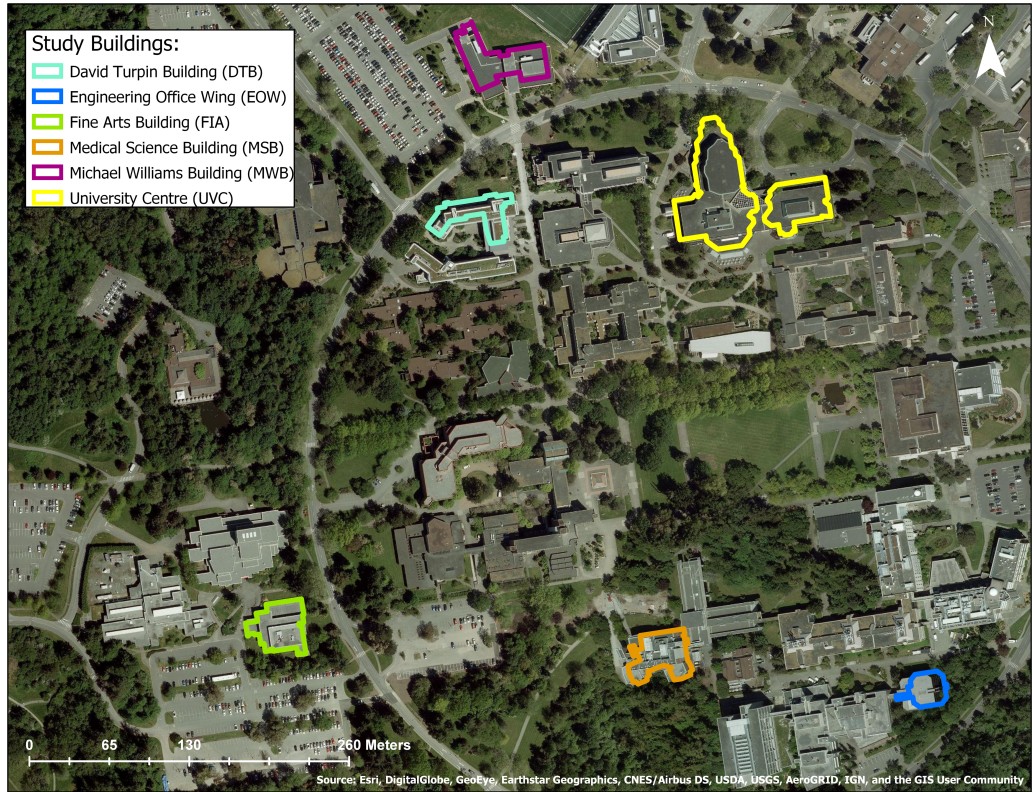

**Figure 2** **Aerial view of study site at the University of Victoria, Victoria, British Columbia Canada.** Six study buildings chosen using a stratified random design are outlined in color. Source: Esri, Digital Globe, GeoEye, Earthstar Geographics.

window collision evidence included feather smears on windows, or stunned live birds found adjacent to building façades and taken to rehabilitation facilities, *i.e.,* the survival outcome of these collision events was unknown. We assumed that two pieces of evidence represented different individuals only if they were detected at least two meters apart (*e.g.,* if a carcass was found within two meters of a feather smear, only one collision was counted). To improve searcher efficiency, vegetation within the two meters survey zone was cleared by campus grounds staff prior to the commencement of surveys. Labor shortages related to the COVID-19 pandemic prevented trimming of the small amount of vegetation that regrew between our third and fourth survey year. However, this did not significantly affect searcher efficiency, as confirmed by analysis of our searcher efficiency trials (Table S1).

### Carcass collection, storage, and identification

Bird carcasses and feather piles found by surveyors were collected and frozen for future confirmation of species identification, authorized under scientific permits issued by Environment and Climate Change Canada for possession of migratory bird carcasses and feathers (SC-BC-SC-BC-2019-0012SAL, SC-BC-SC-BC-2020-0012SAL, SC-BC-SC-BC-2021-0012SAL and SC-BC-SC-BC-2022-0012SAL). All feather piles were later independently identified to the lowest level of classification possible (species, genus, or

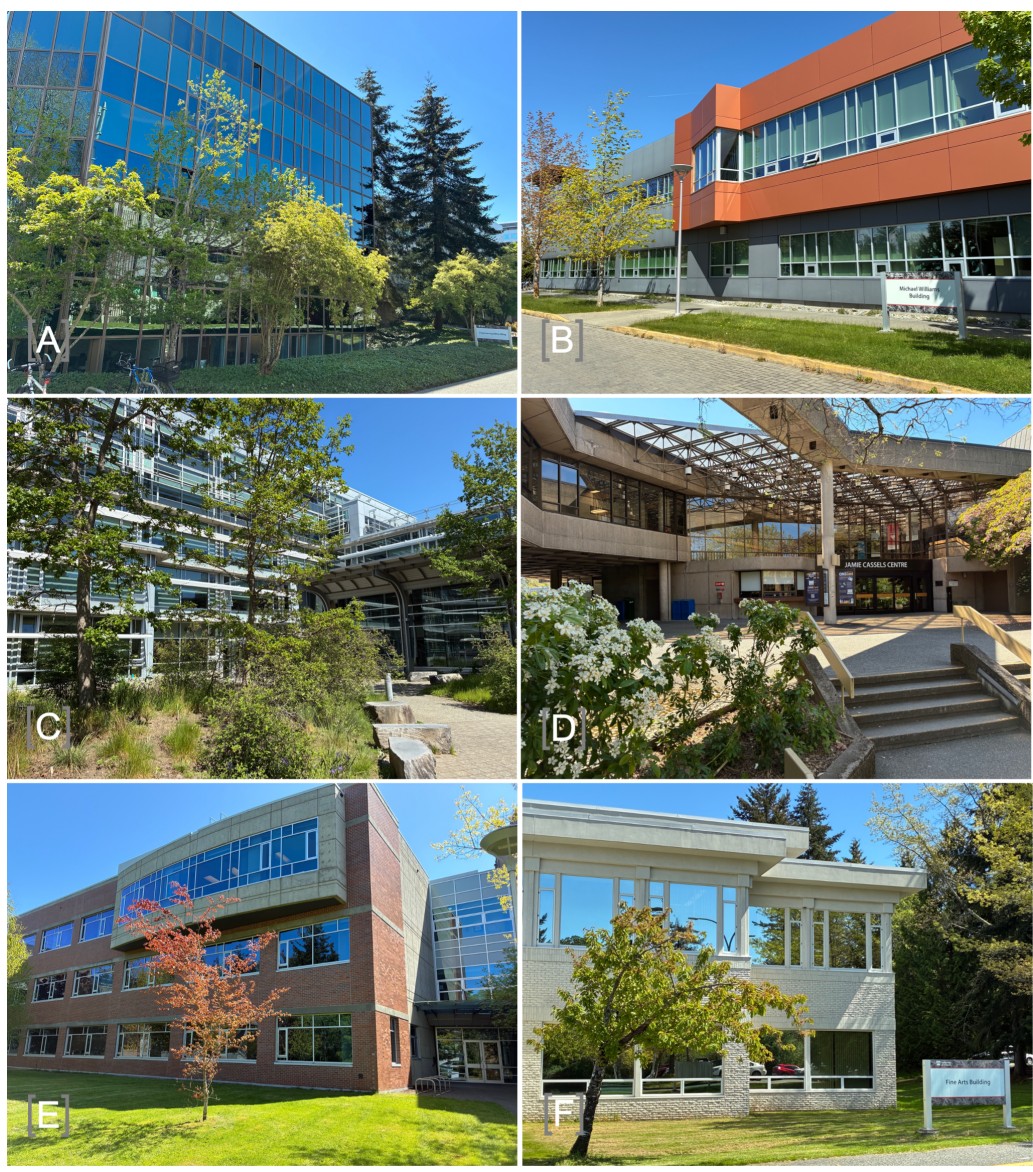

**Figure 3** **Study buildings at the University of Victoria, British Columbia, Canada.** Buildings were chosen using a stratified random design. (A) Engineering Office Wing; (B) Michael Williams Building; (C) David Turpin Building A; (D) University Centre (now Jamie Cassels Centre); (E) Medical Sciences Building; (F) Fine Arts Building.

family) by authors (RG and KLD) and two other experts in feather identification, using the available literature (*Scott & McFarland, 2010*; *Piranga, 2015*; *US Fish Wildlife Service Forensics Lab, 2018*), and feather samples from carcasses of known identity. Challenging identifications, *i.e.,* those for which independently derived identifications did not agree, were assigned "unknown ID".

### Assessment of carcass persistence and searcher efficiency

To assess the efficacy of searchers in finding carcasses that are present, and the persistence of carcasses that could be removed by scavengers or humans prior to surveys (*i.e.,* overall likelihood of surveyors detecting a window-killed bird), we conducted searcher efficiency (SE) and carcass persistence (CP) trials using previously window-killed birds. Eighteen trials were performed in each of the fall 2019 and 2020 study periods (three per building per season), and twelve trials were performed in each of late winter 2020, 2021, and 2022, and early winter 2021 (two per building per season) for a total of 84 SE and CP trials.

Trials were carried out by an observer who did not participate in collision monitoring surveys, without alerting the two bird collision observers to the timing or location of trials. We thawed trial carcasses overnight and clipped their entire hallux so they were identifiable as trial carcasses. These were deployed in a manner that mimicked the fall of a bird from a window, within two meters of randomly selected façades of study buildings, starting at approximately 8:00 on the initial carcass placement day (Day 1). Carcasses were then checked starting at 12:00 and in late afternoon (17:00 in fall and 16:00 in winter) on Day 1, 8:00 a.m., 12:00 noon and late afternoon on Day 2, at 12:00 noon and late afternoon on Day 3 and late afternoon on Days 4 and Day 5, or until trial carcasses or remains were no longer present. Monitoring ended on Day 5. Following scheduled collision monitoring surveys, collision mortality data files were reviewed to ensure trial carcasses were not recorded as collision events. The number of hours from carcass placement until it was found by at least one collision monitoring surveyor was recorded for SE model input (or '0' if trial carcasses were never found). We did not assess inter-individual searcher efficiency, as our model corrects for biases due to carcasses not being located by at least one surveyor. The number of hours from carcass placement to the midpoint between trial checks when the carcass was present and when it was discovered to have disappeared without a trace was recorded and used as CP model input.

### Bird counts for abundance and vulnerability estimates

To assess bird species' abundance on campus relative to individuals killed in window collisions, we conducted five-minute point counts at each study building twice per week through each 21-day survey period (*i.e.,* six point counts per building for each survey period). For each count, the observer was positioned immediately adjacent to the building façade, mid-way along the façade. The sampled area consisted of a semicircle with a 50 m radius, centered at the midpoint of the façade. All birds seen or heard within the semicircle or perched on the edge of the façade (*i.e.,* at distance 0) were counted. To reduce observer effects in our bird counts (*Farmer, Leonard & Horn, 2012*), all point counts were performed by a single observer in fall 2019, and all point counts for the subsequent five seasons were performed by a second single observer.

### Banding station data as independent data source for species abundance estimation

To account for detection probability issues, particularly of species that are quiet or otherwise cryptic during fall migration, we used mist net capture data collected from September 1 to October 18 in 2019 and 2020 from a migration monitoring station (Rocky Point Bird

Observatory; RPBO) situated approximately 30 km southwest of the University of Victoria, on the unceded, traditional territories of the Scia'new and T'Sou-ke First Nations and other ləkʷəŋən and W̱SÁNEĆ Peoples (*NativeLand Digital, 2025*; *Rocky Point Bird Observatory, 2025*), as an independent source of regional abundance data. The banding station is located within the same Coastal Douglas-fir (*Pseudotsuga menziesii*) ecological zone as the University of Victoria, but within a mix of old- and second-growth forest and open fields, with virtually no buildings, and restricted access to the public. The observatory's standardized sampling design consisted of 13 mist nets deployed for a maximum of six hours each day (78 net hours/day). Mist net capture data were summarized as the number of individuals of each species captured per mist net per day.

## Data analysis

### Correcting biases due to imperfect carcass detection and carcass removal by scavengers and humans in bird-window collision and mortality estimates

We used the Generalized Mortality Estimator (*GenEst*) R package (*Dalthorp et al., 2018*) to estimate total collision mortality for our six study buildings during each of the six 21-day surveyed periods. Estimates were corrected for imperfect carcass detection probability by searchers (SE; see Methods) and carcass removal by scavengers or people (CP; see Methods). We also generated total collision estimates using all collision evidence (including feather smears and live bird collisions) as an indication of potential total collision mortality, *i.e.,* the outcome for birds that struck buildings and left a feather smear or were stunned and flew away was unknown, but these collisions may have resulted in mortality or sublethal effects.

For both CP and SE estimates, we fit a null model and combinations of models that allowed both parameters of SE (intercept and slope) or CP (location and scale) to vary by season. For the CP models, we also fit different models using exponential, Weibull, lognormal, and loglogistic distributions to determine best fit for our data. Models were ranked by the Akaike Information Criteria (AIC) and the model with the lower value was selected (Tables S2 and S3). Models with AICc < 2 were considered well supported by our data. We then used the best models' CP and SE estimates to produce bias-corrected estimates of collision mortality by applying the *estM* function of *GenEst* to collision observations, and collision monitoring schedule to propagate uncertainty of estimates using parametric bootstrap methods.

### Campus-wide collision mortality estimates across fall migration and winter

To estimate average mortality and collision numbers that occur campus-wide at the University of Victoria during the peak fall migratory period and the wintering period, we extrapolated our estimates by propagating the uncertainty to the total number of buildings on campus and to the total number of days in each sampled season. Specifically, we extrapolated the season-specific estimated collision mortality and collision counts and associated uncertainty from our 21-day seasonal study periods from our stratified random sample of six buildings to: (1) the 31-day peak fall migratory period between September 15 and October 15 according to the nearby Rocky Point Bird Observatory banding station records (*Leckie, 2008*); (2) the 67-day early winter period between October

16 and December 21 (winter solstice in the Northern Hemisphere) and (3) the 51-day late winter period between December 22 and February 10 (our last day of surveys) to derive an estimate of collision mortality during the sampled fall and winter non-breeding seasons on campus (September 15 to February 10) for our six study buildings; and (4) to the 45 additional institutional buildings on the University of Victoria campus, omitting windowless utility and storage buildings, and houses.

### Species vulnerability to collisions

To estimate the per-capita vulnerability to collisions, we followed an approach similar to *Arnold & Zink (2011)*, using the residuals of a $log_{10} - log_{10}$ regression (Eq. (1)) of estimated total abundance ($N_k^{Total}$) on the number of collisions ($y_k$). Species with positive residuals from the regression are more vulnerable to collisions than average species (*i.e.,* they have more collisions than expected after controlling for abundance), and species with negative residuals are generally less vulnerable to collisions than others. We limited our inference to species for which the 95% credible interval (CI) of the vulnerability measure (residual) excluded 0. We transformed both the total abundance and the number of collisions using $log_{10}$ to respect the non-negative scale of each value, and we added 1 to the number of collisions so that we could include species that we had detected *via* our local abundance data, but for which we had zero recorded collisions.

$$\log_{10}(y_k + 1) \sim Normal(\mu_k, \tau),$$
$$\mu_k = \theta_0 + \theta_1 * \log_{10}(N_k^{Total}). \tag{1}$$

We fit this linear regression within a hierarchical Bayesian model that simultaneously estimated the total abundance using our point count data, or for the fall migration season using both point count data and local mist net capture data. Combining the full analysis into a single model allowed us to account for and propagate the uncertainty in the number of collisions recorded, and the estimated total abundance, into our final estimate of vulnerability. We conducted this vulnerability analysis separately for each season to account for the seasonal changes in the community of birds that may be exposed to collision risk in our study area, although for simplicity, we have not included the season-specific indexing in the equations.

For the fall migration season, the model integrated our replicated point count data from the University of Victoria with the mist net capture data from the RPBO banding station, to estimate the total number of individuals on the campus ($N_k^{Total}$, the predictor in the vulnerability calculation of Eq. (1)). For the four winter survey seasons (one early winter, three late winters), the model used only our University of Victoria point count data. The total number of individuals was estimated as a derived parameter, based on a re-scaling of the mean expected counts within the surveyed area of each point count.

Our model applied an N-mixture approach (*Royle, 2004*) where for each point count ($i$), we estimated the species ($k$) and site ($i$) abundance ($N_{ik}^{Point}$) as:

$$N_{ki}^{Point} \sim Poisson(\lambda_k * area_i), \tag{2}$$

where the $area_i$ is the sampled area of each point count ($0.004 \, \text{km}^2$), which is given as data. For each mist net sampling $n$, we also estimated the species-specific abundance $N_{kn}^{Band}$ as:

$$N_{kn}^{Band} \sim Poisson(\lambda_k * area\_net_n), \tag{3}$$

where $area\_net_n$ is the hypothetical area that each net is sampling and is a variable estimated by the model. By sharing a $\lambda_k$ parameter with both point counts and mist net capture data, we were able to correct the abundance estimates for cryptic species that are captured in mist nets and detected in collision surveys but not detected on point counts.

For the point count observation model, we defined the counts of each species ($k$), point ($i$), and visit ($j$), $counts_{kij}$ as following a binomial distribution, with sample size $N_{ki}^{Point}$ and species-specific detection probability $p_k^{Point}$:

$$counts_{kij} \sim Binomial\left(N_{ki}^{Point}, p_k^{Point}\right). \tag{4}$$

We assumed that the detection probability varied among species and that it could be influenced by flocking behavior (*i.e.*, individuals from a flocking species are more likely to be detected when compared with non-flocking species). We modeled $p_k^{Point}$ as:

$$logit\left(p_k^{Point}\right) = \alpha_k + \beta_{[flock_k]}, \tag{5}$$

setting $flock_k$ as 0 for non-flocking species and 1 for flocking species. For the mist net capture data observation model, we defined the counts of each species $k$, net $n$, and visit $j$, $counts\_band_{knj}$ as following a Binomial distribution, with sample size $N_{kn}^{Band}$ and detection probability $p_k^{Band}$, which is weighted by the effort of each visit, $effort_j$:

$$counts\_band_{knj} \sim Binomial\left(N_{kn}^{Band}, p_k^{Band} * effort_j\right). \tag{6}$$

For effort, we calculated the proportion of effort of a given visit $j$ relative to the maximum effort (the maximum effort at RPBO is 78 mist net hours per visit).

Finally, the total abundance for each species $N_k$ was defined as a draw from a Poisson distribution with a species-specific mean $\lambda_k$:

$$N_k^{Total} \sim Poisson(\lambda_k), \tag{7}$$

where $\lambda_k$ is the relative abundance of each species $k$.

The integrated abundance model described above was applied for the fall data, while the early and late winter abundance estimates were performed using a simpler model, which uses only point count data (*i.e.,* same model described, excluding Eqs. (3) and (6)). Models were fit in a Bayesian framework using JAGS in an interface with R (*R Core Team, 2025*) using the *jagsUI* package (*Kellner, 2015*; see GitHub page for entire R code). Estimates were produced by a Markov Chain Monte Carlo (MCMC) algorithm with four chains, 200,000 iterations, burn-in phase of 100,000, and thinned to every 100 iterations. We used vague priors for all parameters and assessed the convergence of the models by checking the MCMC trace plots and using the Gelman–Rubin statistic ($\hat{R} < 1.1$; *Gelman et al., 2013*).

## RESULTS

### Raw mortality and collision data

We detected a total of 137 collision mortalities across all six 21-day study periods at our six study buildings (average of 22.8 birds killed per building over 126 surveyed days). Of these, 69 occurred in fall 2019, and 44 in fall 2020 (two-year average of 9.4 birds killed per building). We observed 11 fatalities in early winter 2021, and five, three, and five fatalities in late winter 2020, 2021, and 2022, respectively, for an average of 1.8 birds killed per building in our single early winter period, and a three-year average of 0.7 birds killed per building in late winter.

When including all collision evidence (carcasses, feather smears and stunned birds) collected during each 21-day study period, we recorded 315 collision events in total (average of 52.5 per building over 126 surveyed days). This included 117 events in fall 2019, and 108 in fall 2020 (two-year average of 18.8 collisions per building). In early winter 2021, we observed 50 collision events, and in late winter we observed six, 27, and seven in 2020, 2021, and 2022 respectively: an average of 8.3 collisions per building in our single early winter period, and a three-year average of 2.2 collisions per building in late winter.

### Carcass persistence and searcher efficiency

Based on the AIC model selection, the null models were among the best ranked models (AICc < 2) for both CP (Table S2) and SE (Table S3) analysis. Therefore, we used the null model estimates for the bias-corrected mortality estimates in our study.

### Bias-corrected mortality estimates for six study buildings during 21-day survey periods

After accounting for carcasses missed by searchers and removed by scavengers or humans prior to daily surveys, we estimated that 180 birds (95% CI [160–206]) were killed at our six study buildings (average of 30 fatal bird collisions per building) over our six 21-day study periods during fall migration, in early winter, and in late winter at the University of Victoria. Mortality during the fall migratory periods was significantly higher than during both early and late winter, but early winter mortality was significantly greater than late winter mortality (Fig. 4A). When accounting for all collision evidence, including feather smears and live stunned birds, early winter collision totals were comparable to collision totals during the fall 2019 study period (Fig. 4B).

### Campus-wide mortality estimates

We extrapolated our collision mortality estimates spatially and temporally and propagated uncertainty to estimate that on average, 1,095 collision mortalities (95% CI [885–1,342]) occur across all 51 institutional buildings at the University of Victoria in the 149 days between September 15 and February 10 (Fig. 5A). When we account for the total number of collisions (including live birds and feather smears), the estimated number of collisions is more than double, with an average of 2,651 collisions (95% CI [2,267–3,077]) (Fig. 5B).

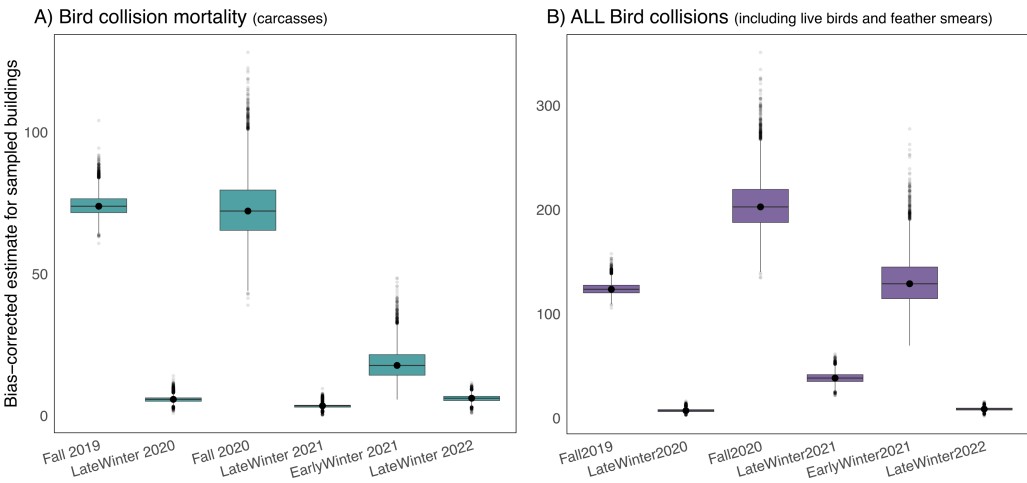

**Figure 4** **Seasonal bird mortality and bird collision estimates for six study buildings at the University of Victoria, British Columbia, Canada.** Buildings were chosen *via* stratified random design, and estimates are corrected for biases due to searchers missing carcasses and removal of carcasses by scavengers or humans. (A) Depicts bird mortality estimates derived from carcasses and feather piles (known outcome) and (B) depicts all collision evidence including feather smears and live birds (known and unknown outcome).

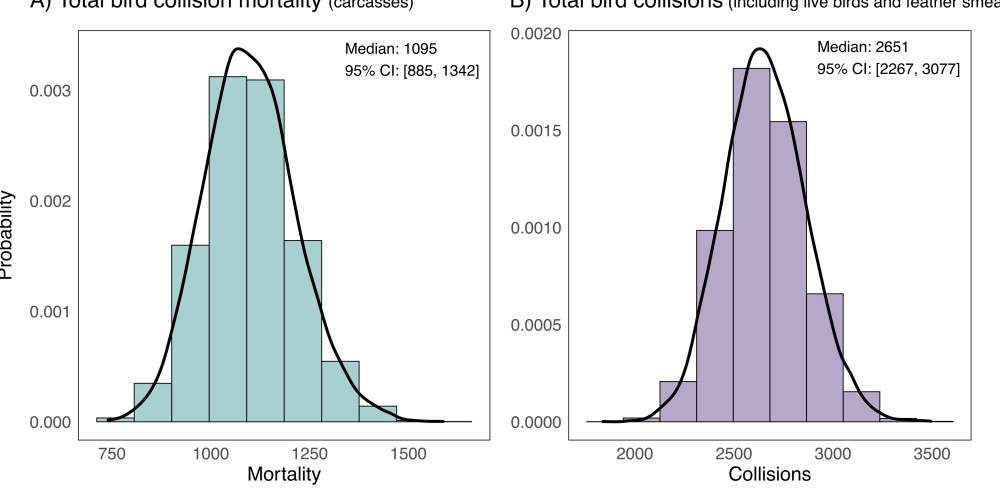

**Figure 5** **Estimates of campus-wide bird mortality and all bird collisions between September 15 and February 10 (fall and winter periods) at 51 institutional buildings at the University of Victoria, British Columbia, Canada.** Estimates are for 51 institutional buildings on campus and are corrected for biases due to imperfect detection of bird collision evidence from searchers missing carcasses and removal of carcasses by scavengers or humans. Estimates are extrapolated from bias-corrected average estimates of two fall 21-day sampling periods, one early winter 21-day period and three late winter 21-day sampling periods at a stratified random sample of 6 study buildings on campus. (A) Depicts bird mortality estimates derived from carcasses and feather piles (known outcome) and (B) depicts all collision evidence including feather smears and live birds (unknown outcome).

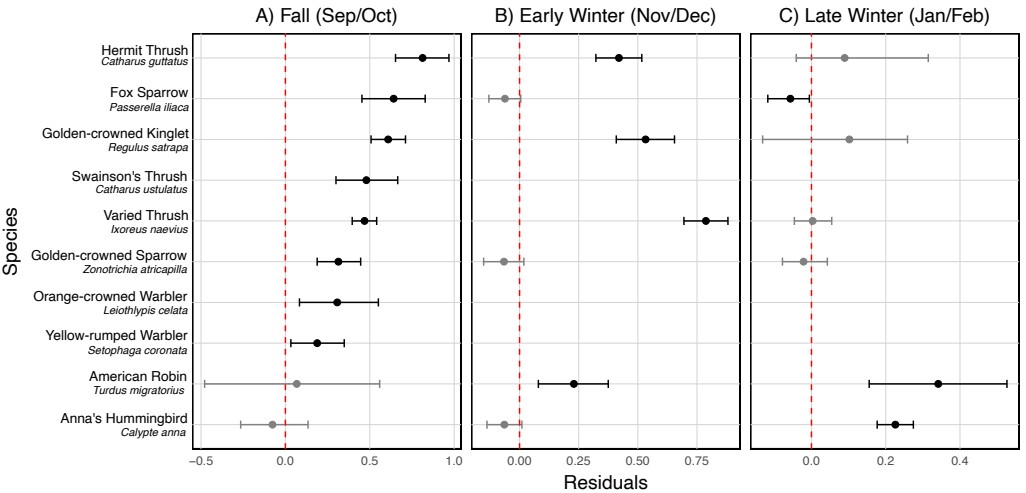

**Figure 6** **Residuals for the most vulnerable species to bird-window collisions at the University of Victoria, British Columbia, Canada across three periods of the non-breeding season.** Results are shown for (A) fall (September/October), (B) early winter (November/December), and (C) late winter (January/February). Each point represents the mean residual, and horizontal error bars indicate the 95% credible interval (CI). Species are considered vulnerable if they have positive residuals with 95% CIs that do not intersect zero (indicated by the vertical dashed line). The species shown are those identified as vulnerable in at least one of the three periods.

## Collision data by taxonomic group, corrected abundances and species vulnerability to collisions

A total of 21 species were detected in collision evidence across all fall and winter study periods, including thrushes (Turdidae; 53 collisions), New World sparrows (Passerellidae; 38 collisions), kinglets (*Regulus* spp.; 20 collisions), and New World warblers (Parulidae; 16 collisions) (Fig. S1). Corrected species' abundance estimates used integrated data from point counts conducted at the University of Victoria and mist net capture data from RPBO (fall only), and incorporated a behavioral factor related to detectability (*i.e.,* flocking; all seasons). Corrected abundance estimates improved our species vulnerability analyses, particularly for species that were present on campus and recorded as collision mortalities, but were not detected on point counts (*e.g.,* Swainson's thrush (*Catharus ustulatus*) and Yellow warbler (*Setophaga petechia*); Tables S4 and S5).

In fall, Golden-crowned kinglet (*Regulus satrapa*), Yellow-rumped warbler (*Setophaga coronata*), Orange-crowned warbler (*Leiothlypis celata*), Hermit thrush (*Catharus guttatus)*, Swainson's thrush, Varied thrush (*Ixoreus naevius*), Fox sparrow (*Passerella iliaca*), and Golden-crowned sparrow (*Zonotrichia atricapilla*) were disproportionately vulnerable to collisions when accounting for their abundance (Fig. 6A, Fig. S1). Golden-crowned kinglet, Hermit thrush, American robin (*Turdus migratorius*), and Varied thrush were particularly vulnerable to collisions during early winter (Fig. 6B, Fig. S1), while Anna's hummingbird (*Calypte anna*) and American robin were the most vulnerable during late winter (Fig. 6C, Fig. S1).

## DISCUSSION

### Collision mortality estimates and comparisons

Collision mortality was highest at our study site during the fall migratory period, coinciding with maximum numbers of fall migrants moving through the region (*Leckie, 2008*). A peak in collision mortality during migration is a consistent finding across many collision studies in temperate zones of North America (*Hager et al., 2008*; *Borden, Lockhart & Jones, 2010*; *Riding, O'Connell & Loss, 2021*), including one previous study that focused on two buildings with a history of collisions at the University of Victoria, our study site (*Hiemstra, Dlabola & O'Brien, 2020*). However, our average annual fall raw mortality rate of 9.4 birds killed per building was more than eight times higher than the average of 1.2 birds killed per building from total mortality counts at 40 primarily campus sites across the U.S. and Canada using the same standardized collision survey methodology and study duration (21 days) in fall (*Hager et al., 2017*), and was double the fall average of 4.7 birds killed per building during the first 21 days of surveys at the University of British Columbia (hereafter UBC) in Vancouver, Canada (*De Groot et al., 2021*). Although fall migration traffic along the Pacific coast is generally considered to be lower compared to eastern North America (*Dokter et al., 2018*), the shape of the western North American coastline likely funnels birds migrating south from Alaska, the Yukon Territory, and north-coastal B.C., through our study location. As with other stopover sites adjacent to large water bodies, birds may spend more time in the region, waiting for good weather prior to flights across the Juan de Fuca Strait to the Pacific mainland coast of the U.S. (see Fig. 1). In addition, some molt migrant species, such as Swainson's thrush, spend significant periods in northern migratory stop-over sites (average of 47 +/−19 days in Montreal, Canada for this species; *Morales et al., 2022*). Frugivorous fall migrants preferentially choose shrubby areas with high fruit abundance (*Mudrzynski & Norment, 2013*; *Smith et al., 2015*; *Poirier, Elliott & Frei, 2024*), and migrants may select habitat at our study site due to a higher availability of energetically-dense native fruits (*Smith, De Sando & Pagano, 2013*; *Blanc-Benigeri et al., 2024*), relative to adjacent suburban vegetation and forest fragments. These factors, combined with typical high-risk features such as high proportion of glass on buildings, large façades, and presence of large trees and shrubs reflecting in building glass, creates conditions for high collision mortality (L Blight, K De Groot, J Osborne, R Golat, unpublished data, 2019–2022; *Klem et al., 2009*; *Cusa, Jackson & Mesure, 2015*; *Hager et al., 2017*; *Riding, O'Connell & Loss, 2020*).

Collision mortality in early winter was significantly higher than mortality in late winter at our study site. When accounting for all collision evidence, including feather smears that indicate a collision has occurred, and stunned birds for which survival outcome was unknown, the early winter period had similar rates of collision to the fall migratory period. Sustained high collision rates in early winter (*i.e.,* late November to early December) may be due to new arrivals to our low elevation study area, as a result of facultative altitudinal migration of montane breeding species or individuals (*Hahn et al., 2004*; *Swanson, Ingold & Galati, 2020*), following significant snowfall and declining food resources in high elevations (*Campbell et al., 1997*). Newly arriving birds to a given location are thought to be more vulnerable to collisions as they seek food and shelter within unfamiliar environments
(*Nemes et al., 2023*) and may also be more likely to collide with buildings during snowstorms and other inclement weather (*Newton, 2007*; *Loss et al., 2020*). At an institutional building in Golden Gate Park, San Francisco, *Kahle, Flannery & Dumbacher (2016)* noted an influx of early winter arrivals in November, and collision mortality during that month was comparable to September mortality. As with our study site, San Francisco is situated within the Pacific Coast Ranges, which extend from Alaska to Central Mexico. Elevational movements are rarely considered in the context of temporal collision risk, likely due to the spatial bias of collision studies that are predominantly conducted in less mountainous regions in eastern and central U.S. and Canada. A caveat to this result is that unlike our fall migratory and late winter periods, we studied the early winter period during a single year (2021). Inter-annual variation in collision mortality may occur (*De Groot et al., 2022*) and we also cannot rule out non-seasonal changes in site or other conditions that may have affected collision mortality results.

Contrary to our expectations, average annual raw mortality in late winter (0.7 collision deaths per building) was over three times lower at our study site compared to average mortality over the same number of days in late winter at the Vancouver, B.C. campus of UBC (2.5 collision deaths per building for the first 21 surveyed days of late winter in that study, corresponding to our study's dates; *De Groot et al., 2021*). Due to mild winter temperatures, Victoria and Vancouver support high densities of overwintering passerines (*Meehan et al., 2022*). However, despite their geographical proximity, there are key differences in microclimate between the two regions, *i.e.,* the University of Victoria is situated within a rain shadow in the maritime Coastal Douglas-fir biogeoclimatic zone, while UBC is within the rainforest of the Coastal Western Hemlock (*Tsuga heterophylla*) biogeoclimatic zone (*Meidinger & Pojar, 1991*). Differences in moisture may affect the temporal availability of food resources and thus bird distribution and movements across the winter months; for example, moisture was found to play a key role in driving food abundance and movement patterns of Wood thrush (*Hylocichla mustelina*) that spent the boreal winter period in Belize (*Stanley et al., 2021*). Additionally, while there are non-native plants on the University of Victoria campus, the predominantly native vegetation landcover may supply more temporally limited food resources, particularly for seasonally frugivorous species that were most vulnerable to collisions in winter, as indicated below.

## Campus-wide estimates of bird collisions across fall and winter

We estimate that on average 885–1,342 (median = 1,095) birds are killed at the University of Victoria each fall and winter season between September 15 and February 10: an average of 21.5 birds killed per building across the 149-day period. We did not extrapolate our estimates to the remainder of the year; however, it is notable that the per building mortality rate for approximately 40% of the year is similar to the average mortality rate of 21.7 collision deaths per building over a full year period used to calculate annual collision mortality at mid-rise buildings in Canada (*Machtans, Wedeles & Bayne, 2013*). The Canada-wide estimate assumes minimal winter collisions at mid-rise buildings in the country (*Calvert et al., 2013*), an assumption that is not supported by our research and that of *De Groot et al. (2021)* at Pacific coastal locations, when accounting for mortality across the full winter

period (*Zulian et al., 2023*). Our estimate of collision occurrence campus-wide for the fall and winter periods, which includes stunned birds and evidence from feather smears, is considerably higher (2,267–3,077; median = 2,651; 52 birds killed per building in fall and winter). Given recent findings that 60% of stunned birds brought into care succumb to their injuries and that 50% or more collisions leave no visible evidence (*Samuels et al., 2022*; *Kornreich et al., 2024*; *Klem, Saenger & Brogle, 2024*), collision mortality at our study site in fall and winter may be closer to the latter estimates.

## Species-specific differences in vulnerability to collisions

We assessed species vulnerability to collisions over each seasonal period by determining whether collision mortality for a given species could be predicted by local abundance alone. Some species such as Swainson's thrush are cryptic during fall migration and were not detected in our local fall point counts but were clearly present at the site because they were killed by colliding with our study buildings. This indicated that point counts alone were not a reliable measure of abundance for all species during fall migration. We therefore devised an approach to integrate data from our building-level point counts with mist net capture data from a nearby bird banding station to better estimate species' numbers. We found that species-specific collision vulnerability varied by season, but Golden-crowned kinglet, Hermit thrush, Varied thrush, and American robin showed above-average vulnerability to collision morality over more than one period (fall, early or late winter; Fig. 6). Hermit thrushes are short-distance latitudinal and elevational migrants, whereas the other species are partial migrants, *i.e.,* some individuals of each species breed locally (within 5 km of the study site), but are also joined in the non-breeding season by latitudinal and elevational migrant individuals (*Chapman et al., 2011*). Swainson's thrush was significantly more vulnerable to collisions during fall migration (Fig. 6) and is added to the list of highlighted vulnerable species because this species, being the sole obligate long-distant migrant (*Mack & Yong, 2020*) showing above average vulnerability, is not present at our study site in the early and late winter periods.

### Breeding habitat association, non-breeding season diet and bird collision vulnerability

There are two main similarities among the most vulnerable species to collisions at our study site. All are forest-breeding birds, with the American robin inhabiting the broadest range of forest and parkland conditions (*Dellinger et al., 2020*; *George, 2020*; *Swanson, Ingold & Galati, 2020*; *Vanderhoff et al., 2020*). Forest birds are considered particularly prone to collisions around buildings surrounded by a lot of urban tree cover (*Cusa, Jackson & Mesure, 2015*), due their propensity to make short, high-speed flights through gaps within dense vegetation (*Klem, 2014*; *Samuels et al., 2022*). However, *Tan et al. (2024)* found that forest cover surrounding buildings was a risk factor primarily for non-migrants in Singapore. Moreover, *Riding, O'Connell & Loss (2020)* reported clear differences between the American robin (typically a short-distance or partial migrant) and Swainson's thrush (long-distance migrant) in the degree to which vegetation and land cover variables affected collision mortality in Oklahoma, U.S. This suggests that breeding habitat association

and migratory strategy of species may not reliably predict collision outcomes across all geographic regions and site-specific contexts.

Frugivory is also emerging as a diet-related risk factor for bird-window collisions in cold weather months. The mechanisms associated with this risk are not known, but may be influenced by the consumption of ethanol in fermented fruits (*Kinde et al., 2012*; *Ocampo-Peñuela et al., 2016*; *Brown et al., 2019*; *De Groot et al., 2021*) and/or behavioral factors associated with tracking fruit resources. Golden-crowned kinglets include fruit in their diets during fall migration, but they are not known to preferentially target fruit (*Carlisle et al., 2012*). However, four of our five most vulnerable species are thrushes (family Turdidae) that are highly frugivorous during the non-breeding season (*Dellinger et al., 2020*; *George, 2020*; *Mack & Yong, 2020*; *Vanderhoff et al., 2020*). Frost and other cold temperature events can accelerate fruit fermentation in fall and early winter, potentially causing ethanol toxicity and disorientation in birds that consume them in high abundance (*Eriksson & Nummi, 1983*; *Fitzgerald, Sullivan & Everson, 1990*; *Kinde et al., 2012*), although future studies are needed to confirm whether this phenomenon is widespread and contributes to collision deaths. Many species of birds, including those that track ephemeral resources such as fruits post-breeding, are highly mobile beyond the migratory period in tropical regions (*Faaborg et al., 2010*; *Ruiz-Gutierrez et al., 2016*). The winter movement patterns of the species most vulnerable to collisions at our study site are not well known; however, they may also be nomadic and non-territorial in temperate wintering areas. Large-scale or frequent movements throughout the non-breeding season may result in higher vulnerability to collisions, due to lack of familiarity with threats in continuously new surroundings (*Kahle, Flannery & Dumbacher, 2016*).

Few clear patterns in vulnerability to collision mortality have emerged within higher-order taxonomic groups of passerines. *Nichols et al. (2018)* noted that collision vulnerability varied considerably within the families Parulidae (New World warblers) and Passerellidae (New World sparrows). *Colling et al. (2022)* also found that individual warbler and sparrow species were both among the most vulnerable and the least vulnerable species in their Toronto, Canada-based study. However, some convergence in individual species assessments of vulnerability occurs in studies that include both migratory periods and winter in their analyses. *Kahle, Flannery & Dumbacher (2016)* also found that Swainson's thrush and Hermit thrush were among the 14 species that collided at a rate that was disproportionate to their abundanc during migration, but overall, hummingbirds were the most highly overrepresented in mortality data at their study building on the Pacific coast. Varied thrush, American robin, and Hermit thrush were reported as highly vulnerable to collisions in late winter at the University of British Columbia (*De Groot et al., 2021*). The American robin and Swainson's thrush were among the most numerous species recorded killed by window collisions in 20 and 17 studies respectively, of 40 studies conducted in the Americas (*Basilio, Moreno & Piratelli, 2020*). Swainson's thrush is also the most numerous species by an order of magnitude, comprising over five percent of all records, within a comprehensive published dataset of bird window collisions in the Neotropics (*Piratelli et al., 2025*). However, the latter two data sets do not account for relative abundance. Further patterns may emerge in individual species' vulnerability to collisions, through additional
studies across a wide range of seasons, geographic areas, and urban to rural contexts. Species that are disproportionately vulnerable across multiple stages of their annual cycle and across their range will be more likely to suffer population-level effects of collision mortality.

## Potential conservation consequences of collision mortality

Collision mortality may have population-level effects if this mortality is additive to natural mortality, *i.e.,* if populations are not able to compensate due to density-dependent processes, including increased survival of other individuals in the population, or *via* increased reproduction (*Longcore & Smith, 2013*). *Dokter et al. (2018)* noted lower survivorship of temperate-zone wintering birds, compared to species that migrate to more southern latitudes; therefore, temperate overwintering species rely on high breeding productivity to maintain stable populations. However, *Rosenberg et al. (2019)* reported that species that overwinter in temperate regions had the largest net losses in abundance in North America since the 1970s, indicating that breeding productivity may no longer compensate fully for non-breeding season losses for some species. Based on Breeding Bird Survey data (*Smith et al., 2024*), Hermit thrush and Golden-crowned kinglet have undergone significant ten-year continental-, Canada- and B.C.-wide (subnational) population declines, with the steepest declines occurring in western and northern regions, to the degree that they may qualify for assessment as Threatened under national criteria in Canada (*The Committee on the Status and Endangered Wildlife in Canada, 2021*). Varied thrush populations have declined significantly over the past ten years (*Meehan et al., 2022*; *Smith et al., 2024*) and over the long-term (>50 years) along the Pacific coast of B.C., Oregon and California (*Smith et al., 2024*), coinciding with the high-density breeding range of the two coastal subspecies (*George, 2020*). However, lower-density far-north populations of the primarily interior-breeding subspecies appear to be stable (*Smith et al., 2024*). It is unknown whether fall migrating and overwintering Varied thrushes at our study site are individuals from primarily coastal or interior subspecies. Swainson's thrush is declining significantly within the range occupied by the Pacific coastal subspecies, which likely migrates through our study area (*Ruegg, 2008*; *Delmore et al., 2016*; *Smith et al., 2024*). Although population declines may be primarily influenced by loss and degradation of breeding habitat (*Dellinger et al., 2020*; *George, 2020*; *Mack & Yong, 2020*; *Swanson, Ingold & Galati, 2020*) and climate change (*i.e.,* changing environmental conditions; *Koenig & Knops, 2022*), the emergence of persistent declines in some geographic areas in all of these species suggests that breeding season productivity may no longer compensate for all sources of mortality in the non-breeding season; therefore collision mortality could be additive and contributing to population declines. A greater understanding of the strength of migratory connectivity between breeding and non-breeding populations may also clarify the potential links between non-breeding season mortality from bird-window collisions and population trends (*Faaborg et al., 2010*; *Loss et al., 2014*).

## CONCLUSIONS

We found that forest birds and species that switch to a diet of fruit in fall and winter were the most vulnerable to bird-window collisions at our study site, potentially due to behavioral factors associated with forest dwelling and seasonal frugivory. Although collision mortality was generally lower in the winter months compared to the fall migratory period, mortality from this threat continues at campus institutional buildings throughout the boreal winter where it has been studied on the Pacific coast. Collision mitigation efforts should focus on geographic areas with the highest cumulative mortality (*Machtans, Wedeles & Bayne, 2013*) considering multiple stages of the life cycle. Future research should identify specific building and landcover collision risk factors, and seasonal differences in these factors, that affect the species most vulnerable to collisions (*Elmore et al., 2021*) across their life cycle. Such research should prioritize species undergoing steep declines, thus allowing for targeted mitigation measures.

While information on regional differences and species vulnerability to collision mortality should inform prioritization of collision mitigation actions, the absence of irrefutable evidence supporting population-level effects of collision mortality should not impede or delay an acceleration of conservation efforts to mitigate mortality (*Longcore & Smith, 2013*; *Klem, 2015*). Birds connect humans with nature, play important roles in cultural and ecosystem relationships with Indigenous Peoples, and bird-human associations have been evident in Indigenous communities for millennia (*Tidemann & Gosler, 2012*; *Delfino, 2024*). In addition, birds provide multiple ecosystem services, and direct economic benefits to communities (*Schwoerer & Dawson, 2022*; *Gaston, 2022*; *Wild Bird Feeding Institute, 2023*). Evidence-based guidance can support building owners to effectively (and attractively) mitigate bird collisions at existing buildings through application of densely spaced markers, designs, or art on glass, that contrast well with reflections, thereby making glass visible to birds (*FLAP Canada, 2025*; *American Bird Conservancy, 2025*). Cost-effective 'multi-solving' approaches are also available to architects in the building design phase, including the use of building-integrated features that can both reduce operational greenhouse gas emissions of buildings, and reduce bird collision risk (*Canadian Standards Association, 2019*; *FLAP Canada, 2025*).

## ACKNOWLEDGEMENTS

We thank Trevor Bulmer and Iwan Lewylle for assistance with data collection, and Sydney Bliss (Canadian Wildlife Service, CWS) and Kerry Kenwood for assistance in identification of feather piles. Thanks to additional CWS staff: Lauryn Do, Amanda Lu, and Kathleen Moore for GIS support, and to Laurie Wilson for management of scientific permits. We are grateful to the Rocky Point Bird Observatory, particularly Daniel Donnecke, for providing mist net capture data and related information for the years of our study, and to Erin Dlabola for providing contacts from her earlier efforts to assess window collision mortality at high-collision buildings on campus. We thank staff and students at the University of Victoria, in particular Neville Winchester for storing frozen carcasses, David Perry, Brent Maness, Peter Roberts, Tom Downie, and Keith Cascon for help with access to buildings,

and introductions to building occupants, and Peter Roberts and his horticultural crew for clearing overgrown vegetation around study building perimeters. Ildiko Szabo (Beaty Biodiversity Museum, University of British Columbia), and Darren and Claudia Copley (Royal BC Museum and Victoria Natural History Society) contributed carcasses to the carcass persistence and searcher efficiency trials.

### Funding
Funding for this work was provided by Environment and Climate Change Canada, including contracts to authors Louise K. Blight, Jon Osborne, Rebecca Golat, and Viviane Zulian. The funders had no role in study design, data collection and analysis, decision to publish, or preparation of the manuscript.

### Grant Disclosures
The following grant information was disclosed by the authors:
Environment and Climate Change Canada.

### Competing Interests
Louise K. Blight is the sole proprietor and principal of Procellaria Research & Consulting and Jon Osborne is sole proprietor and principal of Sanicle Environmental Consulting. Adam C. Smith, Andrea R. Norris and Krista L. De Groot are employees of Environment and Climate Change Canada.

### Author Contributions
- Viviane Zulian analyzed the data, prepared figures and/or tables, authored or reviewed drafts of the article, curated data and code, and wrote code for data integration, and approved the final draft.
- Louise K. Blight conceived and designed the experiments, performed the experiments, authored or reviewed drafts of the article, reviewed and managed data, supervised field operations and co-led project, and approved the final draft.
- Jon Osborne performed the experiments, prepared figures and/or tables, authored or reviewed drafts of the article, reviewed and managed data, and approved the final draft.
- Adam C. Smith analyzed the data, authored or reviewed drafts of the article, and approved the final draft.
- Andrea R Norris conceived and designed the experiments, analyzed the data, authored or reviewed drafts of the article, and approved the final draft.
- Rebecca Golat performed the experiments, prepared figures and/or tables, and approved the final draft.
- Krista L De Groot conceived and designed the experiments, prepared figures and/or tables, authored or reviewed drafts of the article, managed, contributed methods, and co-led project, and approved the final draft.

## Field Study Permissions

The following information was supplied relating to field study approvals (i.e., approving body and any reference numbers):

Environment and Climate Change Canada granted scientific permits for possession of migratory bird carcasses and feathers (SC-BC-SC-BC-2019-0012SAL, SC-BC-SC-BC-2020-0012SAL, SC-BC-SC-BC-2021-0012SAL and SC-BC-SC-BC-2022-0012SAL).

## Data Availability

Code and raw data are available at GitHub and Zenodo:

https://github.com/vivizulian/BirdCollisionsUniVictoria.

Viviane Zulian. (2025). vivizulian/BirdCollisionsUniVictoria: initial (initial). https://doi.org/10.5281/zenodo.16797374.

## Supplemental Information

Supplemental information for this article can be found online at http://dx.doi.org/10.7717/peerj.19943#supplemental-information.

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
