# Peer review of "Seasonally frugivorous forest birds and window collision fatalities: novel integration of bird counts in fall improves assessment of species vulnerability to collisions"

_PeerJ, doi:10.7717/peerj.19943_

## Round 0.1 · original submission · Minor Revisions

Thank you very much for your manuscript titled “Seasonally frugivorous forest birds and window collision fatalities: novel integration of bird counts in fall improves assessment of species vulnerability to collisions” that you sent to PeerJ.
This study presents valuable and relevant information on university campuses as important sources of bird collision mortality.

As you will see below, the comments of both referees suggest a minor revision before your paper can be published. Given this, I would like to see a minor revision dealing with the comments. I will be happy to accept your article pending further revisions, detailed by the referees.

Reviewer 1 suggests in highlighting the distribution of the species and focusing the analysis by groups and focusing part of the discussion on the ecology of the species. He also makes some notable observations about the manuscript that need to be addressed.

Reviewer 1 comments on the manuscript's grammar and raises several questions about the methodology and experimental design.

Reviewer 2 comments on the way some results are reported and on the use of some concepts. He also makes several grammatical suggestions. He also shares a number of personal experiences related to the topic under study.
Please note that we consider these revisions to be important and your revised manuscript will likely need to be revised again.

**Language Note:** The review process has identified that the English language must be improved. PeerJ can provide language editing services - please contact us at [email protected] for pricing (be sure to provide your manuscript number and title). Alternatively, you should make your own arrangements to improve the language quality and provide details in your response letter. – PeerJ Staff

·

Basic reporting

The main grammar issue was the reporting of dates. Most date values were followed by "th" (usually normal font size, but superscript in one or two places; e.g., line 41), but it was inconsistent with some dates lacking a suffix (e.g., line 187). Just make it consistent across all dates throughout the manuscript.

There were multiple places where authors listed in the text were also noted parenthetically, which I assume is an effect of the citation software used. See line 87 (De Groot et al.), line 553 (Tan et al.), line 607 (Dokter et al.)

Line 124: given that you never again use the terms “super collider” or “super avoider”, I’m wondering if this clause should just be removed? It doesn’t really feel like it adds to what you’re saying.

Lines 186-196: I have minor reservations about the seasonal terminology described. Not that it causes issues with the manuscript, just that it feels inaccurate. Astronomical winter is roughly 22 December to 21 March, so your “late” winter period is in the first half of the season and your “early” winter period is in fall (autumn). Based on the late/early modifiers, I was expecting both periods to be several weeks later.

Lines 206, 281, 309, 475, Fig 4 caption: because "fate" has other denotations, I suggest using "outcome" instead.

Lines 479-480: another potential reference to consider here is Loss et al. 2020 https://doi.org/10.2981/wlb.00623. Although the location and season differ, they do show that some species (i.e., American Woodcock) are more likely to collide during/after snow storms.

Lines 614-615: the abbreviation BC here is presumably British Columbia, but as far as I can recall that abbreviation hasn't been defined in the text, which should be considered for an international audience. I suggest rewriting this as "...ten-year continental, national, and provincial population declines"

Throughout the main you did not use the oxford comma, however it was used in some of the supplementary table captions. Suggest making it consistent.

Experimental design

A big issue for me in this manuscript was the stratification of sample buildings into height categories. There was no context given for why building height should matter, and height categories were never mentioned in the Results or Discussion. You need to address why this stratification was used, and why it may have been preferable to some other structural variable, like proportion of window coverage or building footprint. As currently written, the stratification seems completely pointless.

In lines 178-179 you mention low variation in vegetation coverage, but in line 164-165 you mention both native and non-native vegetation. Was there enough variation in native v. non-native to make an interesting comparison?

Based on my reading of your sampling periods, you didn’t do fall, early winter, and later winter in the same “year” (consecutive periods). I understand that you will use abundance counts to provide indices that account for annual variation, but it makes me wonder about other potential interannual differences that are unrelated to bird abundance. For example, what if the lighting schedule at a building was altered in December 2021, leading to a change in collision rates? Even though season was not the primary cause of the difference, you might capture that as a seasonal difference because you had winter but not fall sampling after that.

Lines 198-202: I have several questions about the survey methodology, but I don't think any are necessarily critical. Presumably, the observers encountered each other about halfway around each building. Did they communicate mid-survey about detected carcasses or was that discouraged? Since you didn't model individual observer efficiency, it may not have mattered. How many total observers were there? Was it the same two for every survey? If observers repeated a survey (e.g., on a different day), did individuals vary direction?

Lines 202-207: you distinguished between fatal collisions (carcasses and feather piles) and collision evidence (smears and stunned birds) and quantified those separately. However, I think there's good reason to separate partially scavenged carcasses (including feather piles) from intact carcasses in analyses because non-intact carcasses have a higher likelihood of being a predator kill rather than a window kill. Anecdotally, I've directly observed several instances of a predator (usually a raptor) capturing a bird and leaving evidence of it in a survey area even though the consumed bird did not collide with a building. Please provide some reasoning for not distinguishing or separating intact and incomplete carcasses.

Lines 210-212: You mostly use "searcher efficiency” in the main text, but “carcass detectability” occurs in this sentence and in the caption of Table S1. Although closely related, those are not the same. Searcher efficiency would be a metric of your observers, or how well they are detecting carcasses that are present. Whereas, carcass detectability would be a metric of the carcass, or how easily it can be detected by an observer. Perhaps this distinction doesn’t matter much, and I’m being too persnickety. However, I’d suggest sticking with “searcher efficiency” as that seems to be what you are measuring and analyzing.

Lines 236-241: I assume carcasses were placed within the 2-meter survey area? Were carcasses placed in any particular manner (e.g., on their dorsal surface), or was the placement haphazard, intending to mimic the fall from a window after a collision?

Lines 243-245: This description sounds like searcher efficiency was modeled collectively rather than individually. In some studies, inter-observer variability has been shown to be quite high. Do account for that potential variation in some way? Or are you just assuming that the variation is relatively low?

Lines 250-256: the description in line 253 ("a radius of 50 meters from the study facade") is ambiguous. So, the count area was 50-m from a focal facade, not the observer? Where was the observer in relation to the focal facade during the count? If the facade was 30 m long, does that mean the count area was ~150 m2 (= 30 m * 50 m)? Or does the count area include go 50 m on either side of the facade (e.g., a bird on top of the building would be counted)?

Lines 295-298: I’m not sure your data are appropriate for extrapolating to this entire period (30 Aug to 15 Mar). For example, in line 188 you mention that your fall sampling period coincides with the peak of migrants for this area. Thus, the abundance of birds and the number of collisions are likely higher during your sampling period than they were earlier or later in the fall migratory period. Thus, you would likely be overestimating abundance and collisions, if using that sampling period to represent the entirety of fall migration. Based on your sampling periods, I'd say you can appropriately estimate for about 15 September to 15 February.

Validity of the findings

I have no experience with Bayesian statistics, so I am unable to verify the validity of the modeling techniques, but other analyses seemed appropriate.

Lines 384-394: I have multiple questions here. First, in lines 386-388 you report fatalities "per building" but in lines 391-393 it’s collisions “per day”. Why are these reported in different units? Why not convert both of these to “per day per building”? Having different units makes comparisons difficult. A superficial reading suggests you had more fatal collisions (9.4) than total collisions (5.4). Second, I don’t see anywhere that you’ve addressed the assumption that each piece of evidence (e.g., carcass, smear, etc.) represents a separate individual. That is, if a single bird leaves one or more smears on a window and then also is counted as a carcass or a stunned bird, how are you accounting for that potential overestimation?

Lines 402-404: To facilitate a comparison of this estimate to your observed count, you should provide values in the same units. By my calculation that would be 137 total observed (1.09 per building per day) versus this estimate of 180 (1.43 per building per day).

Additional comments

Overall, I thought the manuscript was well-reasoned and well-written. I had a couple minor quibbles with terminology and many questions about the methodology that should be clarified.

·

Basic reporting

The authors are to be congratulated for this excellent contribution to the literature on bird-window collisions. The study design and analysis are sound, the objectives clear, the data rigorously collected and curated, and the presentation in this manuscript is well-written. (Minor suggested edits below.)

The data show what the data show, i.e., that college campuses can be important sources of collision mortality, that with meticulous attention to detail in analysis we can tease out some spatiotemporal and taxonomic patterns of interest, but that those patterns aren't universal across the literature. The one generalization that has held up through my decades of attention to this topic is that Dan Klem had it (mostly) figured out in the 1970s: birds are vulnerable to collision with clear and/or reflective glass.

The writing is quite good, with logical flow of information through sections and paragraphs. Here are just specific quibbles:

1) The only clunky bit of writing that caught my eye was in lines 384–394 to begin the Results. There is nothing grammatically problematic here, but I had some difficulty following the jumping around between years and among winter seasons. I'm not sure there is a better way to communicate this, but I encourage the authors to give a little thought to smoothing out this section. This is one of those 10-minute problems: If you can come up with something better in 10 minutes, do it. If you can't then move on, because it's not worth any more than 10 minutes of your attention.
2) Use of the word "habitat" is mostly fine but in some cases is redundant or pedantically misused. For example, "suburban habitat" in line 467 is literally "the complex of conditions that support the occurrence of low-density human habitations nearby higher-density human habitations", but I suspect that the authors' intent was "suburban cover", i.e., "single-family homes embedded in a matrix of impervious surfaces, regularly-mown turfgrasses, and both remnant and ornamental woody vegetation." This is a complex example that can be fixed by switching to something like "suburban cover" or "suburban areas" but should still be defined somewhere. An easier fix is illustrated in line 499: "rainforest habitat" can simply be referred to as "rainforest."
3) Please review entire document for accurate use of hyphen -, en dash –, and em dash ––. For example, in line 404 "21-day" is an approriate use of a hyphen, but in line 403 the value range "160-206" should instead use an en dash, "160–206."
4) There are multiple inconsistencies in the formatting of Literature Cited entries. Please review to make sure binomials are italicized and for consistent use of sentence versus title case.

Figures are clear (your campus is beautiful, incidentally) and add to the reader's understanding.

All data tables are clear and accessible.

Experimental design

In my estimation, the authors have used robust methods in design, implementation, and analysis. Some aspects of their coding and modeling are beyond my ken, and different authors have followed slightly different methods in calculating vulnerability, but the structure and the inputs make sense. If you want to know which species might collide more or less frequently than expected, you need some way to estimate the numbers of those species "available" for collision to interpret the number of individuals of different species found to have collided (analogous to "use" of the windows).

Validity of the findings

This work represents the state of the science in the field and illustrates well how such studies should be undertaken.

With each new study in this field, however, I am struck by how little we can generalize across spatial and temporal scales. Seasonal differences in collision rates vary, the influence of lighting versus vegetation varies, the super-colliders and super-avoiders vary. The only generalizable things across this field are that birds die from colliding with glass and we want to find ways to stop that. Thus, all of these manuscripts end up at the same point: we assume the mortality is additive, we need to do more to retrofit existing structures to decrease mortality, and we need to enshrine better practices in new construction.

Here's another analogy: We noticed that a lot of people were dying in automobile accidents in the latter 20th Century. So we studied that 9 ways to Sunday, and we figured out that seatbelts would really help to solve that problem. But that knowledge alone didn't move the needle. It wasn't until we had standards to require seatbelts in our cars and then when we required people to use them that collision mortality actually decreased. Then we were free to refine further with airbags, anti-lock brakes, etc. This is where I am on this problem: What is our version of seatbelts-in-cars and what are the obstacles to making that happen?

To be clear, I'm not expecting the authors of this manuscript to address those questions. I'm just pointing out that here is another excellent study that, for all its individual and carefully-determined findings, points in the same general direction as all the others.

For a little more interesting perspective, Dan Klem's 1990 paper presented an upper-level estimate of nearly 1 billion birds killed by window collisions in the US, each year. Prior to that, Dick Banks had estimated ~1 window-killed bird per square mile of land, leading to an annual estimate of 3.5 million casualties per year. Klem used US Census data to estimate the number of US buildings and provided a range from his studies of 1–10 casualties per building per year. That's how he got to an upper-level estimate of >900 million, with careful acknowledgement that some buildings would claim 0 victims and some would claim many. The latter could be skyscrapers on major migration routes or residences with bird feeders –– his most deadly residence claimed 33 victims/year.

I got involved in this stuff in 1993 when Klem's 1989 and 1990 papers had come out and Ricky Dunn had just published the FeederWatch analysis that demonstrated Klem's suppositions about high mortality to be common at homes with bird feeders. I was working on something else: low rise buildings in a corporate office park campus with lawns and trees and reflective glass, but no feeders and no skyscrapers. I studied four buildings over the course of one year and developed an average per-building mortality estimate of 29 –– right in line with the estimate reported herein on line 511.

Just switching gears here to conclude, an obvious important part the authors wished to demonstrate in the manuscript is the relationship between frugivory of wintering birds and collisions on their campus. This phenomenon is well known but seldom demonstrated with rigor as here.

I am unsure about the degree to which it has been demonstrated that ethanol from fermented fruits causes birds to become impaired and leads to increased collisions. I am even skeptical about the degree to which "waxwings get drunk" as the evidence I've often seen for these claims is not a specific analysis of blood-alcohol content in the birds, but a bunch of waxwings observed on the ground near fruiting trees and shrubs, with some of them dead and some appearing "drunk". The assumed "impairment" is not necessary to bring about the observation that to me looks like a bunch of waxwings that hit a nearby window and died and some of them survived but are dazed. I suspect the phenomenon has more to do with normal waxwing flocking and flight behavior near buildings (where we humans plant fruiting trees and shrubs) and less to do with literal drunkenness on the birds' part.

Kinde et al. 2012 make a fine case that at least some Cedar Waxwings collected and analyzed after window collision showed evidence of impairment-level ethanol toxicity, but their conjecture that the toxicity *caused* the collisions is in my mind not well established by the data. Thus, unless there is a specific causative analysis in this study between alcohol impairment and reduced capacity to avoid window collisions, I recommend walking back a bit some of the statements in the paragraph beginning in line 561. The frugivory correlation is there; the causation from alcohol impairment is not.

Additional comments

No further comments –– terrific paper!

---

## Round 0.2 · accepted · Accept

After reviewing this revised version of your manuscript, I see that the main comments suggested by the reviewers have been included, while the suggestions not considered are justified in detail. Therefore, I am satisfied with the current version and consider it ready for publication.